Who bears the cost of forest conservation?

Poudyal Mahesh 1
http://orcid.org/0000-0002-5199-3335 Jones Julia P.G. 1 julia.jones@bangor.ac.uk
Rakotonarivo O. Sarobidy 2
http://orcid.org/0000-0002-7426-8152 Hockley Neal 1
http://orcid.org/0000-0002-0083-9872 Gibbons James M. 1
Mandimbiniaina Rina 3
Rasoamanana Alexandra 3
Andrianantenaina Nilsen S. 3
http://orcid.org/0000-0003-1308-2845 Ramamonjisoa Bruno S. 3
1 School of Environment, Natural Resources and Geography, Bangor University , Bangor , UK
2 Biological and Environmental Sciences, University of Stirling , Stirling , UK
3 Département des Eaux et Forêts, Ecole Supérieure des Sciences Agronomiques, Université d’Antananarivo , Antananarivo , Madagascar
Costanza Robert
Electronic publication date: 2018 Jul 5
Publication date: 2018
Volume: 6
Electronic Location ID: e5106
Received 2018 Apr 19; Accepted 2018 Jun 6
Copyright: © 2018 Poudyal et al.
Copyright year: 2018
Copyright holder: Poudyal et al.
License: This is an open access article distributed under the terms of the Creative Commons Attribution License, which permits unrestricted use, distribution, reproduction and adaptation in any medium and for any purpose provided that it is properly attributed. For attribution, the original author(s), title, publication source (PeerJ) and either DOI or URL of the article must be cited.
License URL: https://creativecommons.org/licenses/by/4.0/

Dataset described in: Household economy, forest dependency & opportunity costs of conservation in eastern rainforests of Madagascar 5 23 10 2018 180225 Scientific Data PMC6198750 30351304
Keywords: Sustainable development goals, Human well-being, Opportunity cost, REDD+, Tropical rainforest, Protected areas, Forest conservation, Conservation science, Social and environmental safeguards, Compensation

Funding: Ecosystem Services for Poverty Alleviation Programme NE/K010220/1 Forest and Nature for Society (FONASO) This work was part of the p4ges project (http://www.p4ges.org) funded by the ecosystem services for poverty alleviation programme (grant code (NE/K010220/1). O. Sarobidy Rakotonarivo received European Commission for support through the Forest and Nature for Society (FONASO) joint doctoral programme. The funders had no role in study design, data collection and analysis, decision to publish, or preparation of the manuscript.

==============================
Background

While the importance of conserving ecosystems for sustainable development is widely recognized, it is increasingly evident that despite delivering global benefits, conservation often comes at local cost. Protected areas funded by multilateral lenders have explicit commitments to ensure that those negatively affected are adequately compensated. We make the first comparison of the magnitude and distribution of the local costs of a protected area with the magnitude and distribution of the compensation provided under the World Bank social safeguard policies (Performance Standard 5).

Methods

In the Ankeniheny-Zahamena Corridor (a new protected area and REDD+ pilot project in eastern Madagascar), we used choice experiments to estimate local opportunity costs (n = 453) which we annualized using a range of conservative assumptions concerning discount rates. Detailed surveys covering farm inputs and outputs as well as off-farm income (n = 102) allowed us to explore these opportunity costs as a proportion of local incomes. Intensive review of publically available documents provided estimates of the number of households that received safeguard compensation and the amount spent per household. We carried out a contingent valuation exercise with beneficiaries of this compensation two years after the micro-development projects were implemented (n = 62) to estimate their value as perceived by beneficiaries.

Results

Conservation restrictions result in very significant costs to forest communities. The median net present value of the opportunity cost across households in all sites was US$2,375. When annualized, these costs represent 27–84% of total annual income for median-income households; significantly higher proportionally for poorer households. Although some households have received compensation, we conservatively estimate that more than 50% of eligible households (3,020 households) have not. Given the magnitude of compensation (based both on amount spent and valuation by recipients two years after the compensation was distributed) relative to costs, we argue that no one was fully compensated. Achieving full compensation will require an order of magnitude more than was spent but we suggest that this should be affordable given the global value of forest conservation.

Discussion

By analyzing in unprecedented depth both the local costs of conservation, and the compensation distributed under donor policies, we demonstrate that despite well-intentioned policies, some of the poorest people on the planet are still bearing the cost of forest conservation. Unless significant extra funding is provided by the global beneficiaries of conservation, donors’ social safeguarding requirements will not be met, and forest conservation in developing countries will jeopardize, rather than contribute to, sustainable development goals.

Introduction

Over the past two decades a series of high profile initiatives have highlighted the links between environmental conservation and human well-being (Millennium Ecosystem Assessment, 2005; TEEB, 2010). The UN Sustainable Development Goals (SDGs), which were agreed by the United Nations in 2015, have embedded into international policy the view that ending poverty cannot be achieved without tackling climate change and conserving and restoring ecosystems (Martinez & Mueller, 2015; Gupta & Vegelin, 2016). The loss of tropical forests, for example, has significant societal costs: deforestation and degradation is estimated to contribute 8–10% of global net carbon emissions (Baccini et al., 2012; Tubiello et al., 2015), forests contain highly valued biodiversity (Mittermeier et al., 1998) and contribute to regional hydrological cycles (Lawrence & Vandecar, 2015). For this reason, the sustainable management of forests received special mention in the SDGs (Goal 15). However, the conservation of tropical forests can often have local costs, including physical and economic displacement of people (Balmford & Whitten, 2003; Agrawal & Redford, 2009; Adams, Pressey & Naidoo, 2010; Fisher et al., 2011; Green et al., 2018).

There is longstanding recognition that steps should be taken to ensure that the global good of conservation is not paid for by those least able to bear additional costs. For example, the principle that protected areas should do no harm to local people was established at the World’s Park Congress in 2003 (Pullin et al., 2013). The Convention on Biological Diversity (CBD COP5 Decision V/6) and Aichi Target 11 both require protected area management to be “fair and equitable” (UNEP/CBD, 2000). Increasingly, conservation is funded by major international donors who have explicit commitments to safeguard against negative social impacts. For example, many Reducing Emissions from Deforestation and Forest Degradation (REDD+) pilot projects have been funded by the World Bank (through their Forest Carbon Partnership Facility or other schemes), while major industrial investments such as mines are funding the creation of biodiversity offsets so as to achieve “no net loss” of biodiversity in their operations (Bidaud, Schreckenberg & Jones, 2018). The majority of financial institutions (covering more than 70% of international project finance debt in emerging markets) have committed to the equator principles (Equator Principles Association, 2013). These require projects in countries without robust environmental and social governance to follow the stringent performance standards of the International Finance Corporation; all World Bank projects also follow these standards. Performance Standard five states than where people are displaced, physically or economically, they must be compensated for any losses (IFC, 2012). However, despite these clear commitments to compensation of the local costs of conservation, we know of no example where the magnitude and distribution of the local costs of a conservation project have been estimated and compared with the magnitude and distribution of the compensation delivered.

Madagascar is world renowned for the exceptional biodiversity of its forests making it a focus of conservation attention for decades. In 2003 Madagascar made a high profile announcement that it would triple the size of its protected area network (Gardner et al., 2013); a process which led to the creation of around 100 new protected areas which were finally approved in May 2015 (Republic of Madagascar, 2015a). Madagascar has also been at the vanguard of efforts to pilot the climate mitigation mechanism of REDD+, and REDD+ formed part of the country’s Individual Nationally Determined Contribution to the Paris climate agreement (Republic of Madagascar, 2015b). Madagascar is an extremely poor country with the second highest proportion of its citizens classed as “extremely poor” of any country in the world (World Bank, 2017a). Under both Malagasy law and the requirements of international lenders, new protected areas in Madagascar require an environmental and social impact assessment which identifies who should receive compensation for economic displacement under social safeguard procedures and presents a strategy for delivering such compensation.

It is well understood that human well-being is a multidimensional concept encompassing material, social and subjective components (Woodhouse et al., 2015). Local costs of conservation will not be limited to tangible impacts such as reduced food production: enforced cessation of activities like swidden agriculture may also result in cultural losses. Monetary valuation is one way of bringing together multiple considerations onto a single scale. Discrete Choice Experiments (called choice experiments for the rest of this paper) are a stated preference method which allows estimates of the welfare effects of a project or policy and, crucially, to estimate separately the values of the attributes that characterize that policy (Hanley, Wright & Adamowicz, 1998). While the hypothetical nature of stated preference methods has been long criticized (Hausman, 2012), there have been increasing efforts to tackle various issues with their reliability and validity (Hanley, Wright & Adamowicz, 1998; Louviere, Hensher & Swait, 2000; Louviere, Pihlens & Carson, 2011). With careful design and rigorous field testing, choice experiments can be a useful method for elucidating opportunity costs of land use change or conservation restrictions even in rural areas with limited market integration and low literacy (Kenter et al., 2011; Kaczan, Swallow & Adamowicz, 2013; Nielsen, Jacobsen & Thorsen, 2014; Rakotonarivo, Schaafsma & Hockley, 2016). A particular advantage to using choice experiments for valuing sensitive activities such as illegal forest clearance is that policy impacts are inferred from the trade-offs that respondents make, meaning researchers can avoid asking direct questions about the policy being valued (Rakotonarivo, Schaafsma & Hockley, 2016; Nielsen, Jacobsen & Thorsen, 2014; Moros, Velez & Corbera, 2017).

In this study we use a carefully designed choice experiment (the repeatability and validity of which had been extensively tested: Rakotonarivo et al., 2017, 2018), to estimate the welfare impacts of forest conservation to people surrounding a new protected area and REDD+ pilot project (the Ankeniheny-Zahamena Corridor—CAZ), and contiguous long-established protected areas, in eastern Madagascar (Fig. 1). We estimate the magnitude of local opportunity costs of preventing swidden agriculture on forestlands (commonly cited as the major threat to forest in the region; Styger et al., 2007; Tabor et al., 2017), how these costs are distributed across local households, and the magnitude of these costs in terms of local incomes. We compare our opportunity cost estimates with the estimates produced by the World Bank social safeguard procedures, the money allocated to compensate households, and the value of the compensation that was received (estimated by a contingent valuation exercise).

Figure 1 Map showing the location of the Corridor Ankeniheny Zahamena (CAZ) new protected area in eastern Madagascar.

The location of the four study sites, and the pilot sites are indicated.

Materials and Methods

Study area

The Ankeniheny-Zahamena Corridor (French acronym: CAZ) is a 382,000 ha belt of rainforest in eastern Madagascar linking a number of existing protected areas. It was granted formal status as an IUCN category VI protected area in April 2015. It is also a REDD+ pilot project and received certification to deliver 10 million metric tons of avoided CO2 emissions from Verified Carbon Standard (VCS) over the first 10 year period (Rainforest Alliance, 2013). The habitat is a humid rainforest, and the CAZ and its adjacent long-established protected areas are among the world’s most irreplaceable in terms of biodiversity conservation (Le Saout et al., 2013). The forests are under pressure from expansion of agricultural land (mostly small-scale swidden agriculture), illegal logging and mining (Ratsimbazafy, Harada & Yamamura, 2011; Tabor et al., 2017). Swidden agriculture (the local system of which is known as tavy, or teviala when referring specifically to clearing forestlands) has been the major focus of conservation attention in Madagascar’s eastern rain forests for decades (Scales, 2014). Since colonial times, forest clearance for swidden agriculture has been at times both criminalized and encouraged in Madagascar and enforcement of the current ban is weak, even in long-established protected areas (Kull, 2004).

Around 450 villages surrounding the protected area contain more than 60,000 people who depend primarily on swidden agriculture, and on collecting products from the wild (World Bank, 2012). They are mostly extremely poor and highly vulnerable to economic or environmental shocks (Harvey et al., 2014). As in many other parts of the world (White & Martin, 2002) all forested lands in Madagascar have been formally considered as state land since the colonial era (Horning, 2005). However state ownership is often not recognized as legitimate by local communities and an informal system based on customary rights operates in practice (Antona et al., 2004; Pollini et al., 2014) evolving and adapting in response to state claims and activities (Muttenzer, 2006; Jones et al., 2018).

The CAZ protected area was established with funding provided through the third phase of Madagascar’s environmental plan (PE3; World Bank, 2016). The World Bank requires that all projects carry out social safeguards assessment to identify and mitigate any residual social impacts (Lockwood & Quintela, 2006). The CAZ environmental and social safeguards plan (World Bank, 2012) follows the World Bank guidelines and PE3 framework in laying out the process of identifying and compensating households identified as project affected persons (known commonly as PAPs, but referred to as eligible for compensation in this paper to minimize jargon). Both the PE3 framework and CAZ safeguards plan state that anyone whose sources of income and standard of living would be negatively affected by the restriction of access to the natural resources due to the creation of these protected areas are considered eligible for compensation (ibid.). These documents also specify the need to give special consideration to poor and vulnerable groups; a principle which is central to social safeguards of any World Bank funded project (Hall, 2007). The initial safeguard assessment was conducted in 2010 and identified 2,500 households eligible for compensation in 33 fokontany (the smallest administrative unit in Madagascar). Compensation based on micro-development projects such as improved agriculture, small-scale livestock and beekeeping projects (MEEF/SAPM/CI, 2013) started to be distributed in 2014 (soon after the first phase of fieldwork for this survey was conducted).

Sampling

Following pilot surveys in two areas (see Fig. 1) and reconnaissance visits to a number of others, we purposively selected four sites surrounding the CAZ and adjacent long-established protected areas (see Fig. 1 and Table 1). We selected two sites affected by the new CAZ protected area; one of which received compensation under the World Bank social safeguards scheme (in Ampahitra fokontany) and one which did not (Sahavazina fokontany). Because past exposure to conservation may affect respondents’ stated preferences (Rakotonarivo et al., 2017), we also selected two sites with a long history of conservation under the management of Madagascar National Parks (MNP): Zahamena National Park (which has been mostly under some form of conservation management since 1927; Raboanarielina, 2012), and Mantadia National Park (which was established in 1989; Shyamsundar, 1996). Sites managed by MNP receive development interventions funded through the distribution of 50% of park entrance fees. Available information on the location and size of communities in much of rural Madagascar is very sparse making it difficult to develop a rigorous sampling frame (Poudyal et al., 2016a). Using the available maps as a starting point, we worked with key informants to construct a sketch map showing locations of all villages in each study area. We visited each village, mapped the hamlets and scattered households with the help of key informants and visited the hamlets to map their location with a GPS, confirm the number of households present and ask for information of other households we may have missed. Once we had a complete sampling frame, we selected our sample and an additional 10% to replace households which could not be reached or declined to take part in the survey. Because of the importance of ensuring our sample is fully representative, we devoted substantial time to building the sampling frame; this took up to 30% of total field time in each site.

Table 1 Characteristics of the sites selected for this study, and sample sizes for various surveys.

Site	Fokontany, (commune) district	No. of villages	History of conservation	Enforcement of conservation rules	Compensation provided	HH survey & choice experiment sample	Agri. survey sample	Contingent valuation exercise	
1. New PA with safeguard	Ampahitra (Ambohibary) MORAMANGA	8	Granted temporary status in 2006, formally gazetted in 2015	Weak	Yes (World Bank social safeguards)	102	25	62	
2. New PA (no safeguard)	Sahavazina (Antenina) TOAMASINA II)	7	Granted temporary status–in 2006, formally gazetted in 2015	Very weak	No	95	40	–	
3. Established PA (Zahamena)	Antevibe & Ambodivoangy (Ambodimangavalo) VAVATENINA	7	Long history of conservation (since 1927) on periphery of Zahamena National Park	Relatively strong	Park entry fees shared used to fund local development projects	152	37	–	
4. Established PA (Mantadia)	Volove & Vohibazaha (Ambatavola) MORAMANGA	3	Long history of conservation (since 1989) on periphery of Mantadia National Park	Relatively strong	Park entry fees used to fund local development projects	104	–	–	
	Total	453	102	62	

Data collection

Data for this study was collected through detailed surveys carried out in three phases (see Supplemental Information for copies of all survey instruments in English and Malagasy). First, a general household survey was done with the sample as outlined in Table 1 (July 2014 to March 2015). The survey comprised two sections: (1) socio-economic characteristics of the household including composition, education, wealth indicators (such as land and livestock holdings, house quality and size, access to light); (2) the choice experiment; see below for more details. The household survey and the choice experiment were developed based on preliminary work in the area and extensive piloting (Poudyal et al., 2016b; Rakotonarivo et al., 2017, 2018). We targeted the interviews at household heads but in many cases other household members assisted with recall (particularly in responding to questions about agriculture and collection of wild harvested products). For the second phase (August 2014 to May 2015), a sub-sample of the households who were interviewed in the first phase were selected using stratified random sampling (based on household size and basic information on landholding from our initial survey) for detailed surveys of agricultural input and output, and off farm income. This allowed us to estimate household income from cash and agricultural production (whether sold or consumed), but excluding wild harvested products for subsistence use (many of these products were rarely marketed, making valuation difficult). These surveys were conducted in sites 1, 2 and 3, with sample size of 40–50 households in each site and involved visits to all land farmed by the household to improve recall. We use these estimates of total household incomes to compare with opportunity cost estimates from the choice experiment. In the third phase (May–July 2016), a follow-up survey was carried out with the 62 households in Site 1 (REDD+ safeguard site) who had received compensation under the World Bank social safeguards in 2014. This involved a contingent valuation exercise to estimate their valuation of the micro-development project that they had received.

RM, OSR, NS, AR and up to three additional assistants—all native Malagasy speakers familiar with the dialect of the region—carried out the interviews. MP (basic Malagasy) and NH and JPGJ (fluent in conversational Malagasy) attended a subset of interviews. Pictures showing the fieldwork context are shown in the Supplemental Information. The full data set is archived with ReShare, the UK data services online repository (Poudyal et al., 2016a, 2017). All data, along with the code used in our analysis, is available as a GitHub repository (https://github.com/mpoudyal/cepaper). Research permission was granted by the Ministry of Environment, Ecology and Forests (45/14/MEF/SG/DGP/DCB.SAP/SCB).

The choice experiment

We used a choice experiment to estimate respondents’ willingness-to-accept forest conservation in CAZ, specifically the prevention of swidden agriculture in forested areas. Because there are so few cases of choice experiments being properly validated for use in low income country contexts with rural, low literacy populations (Rakotonarivo, Schaafsma & Hockley, 2016) we invested heavily in refining the choice experiment before rolling it out across the study area. First we explored whether a willingness-to-pay or willingness-to-accept formulation was more suitable. The results showed clearly that willingness-to-accept worked best in this context: it reduced protest responses since it aligned better with local perceptions of (customary) property rights and was more suited to a context where incomes (and therefore ability to pay) are very low (Rakotonarivo et al., 2018). OSR conducted qualitative debriefing interviews to explore the validity of the choice experiment for estimating the opportunity costs of conservation in this context (Rakotonarivo et al., 2017). These were conducted the day after the choice experiment with a sub-sample selected to represent the full range of choice experiment responses (N = 25 from 206 respondents in sites 1 and 4) to examine their decision-making processes.

The choice experiment aimed to assess the net opportunity costs experienced by households prevented from clearing forest for swidden agriculture due to the introduction of conservation restrictions. We asked respondents to choose between a reference level (forest protection is formally lifted and households do not receive any payments or agricultural support) and two experimentally-designed alternatives which varied in our attributes of interest. Choice experiment surveys usually include a status quo option but a status quo option (households’ own current “levels” for each attribute) would vary enormously across respondents and elucidating a status quo alternative would require respondents to reveal their current participation in forest clearance (which is a highly sensitive topic). The attributes were: (i) a monetary attribute (framed as the total development assistance the household would receive); (ii) number of annual instalments over which the household would receive the assistance; (iii) support in-kind for improved rice cultivation; and (iv) forest clearance attribute. The forest clearance attribute had three levels: free clearance (forest protection is lifted), permit for one hectare of clearance, and no clearance (strict enforcement of forest conservation). The attributes and levels (Table S1) were informed by the literature, extensive piloting and three focus group discussions. In particular, levels of the monetary attribute were informed by previous estimates of forest protection opportunity costs in Madagascar and piloting to ensure an adequate level of trading off between this attribute and forest protection (see Rakotonarivo et al., 2017). An example choice card is available in the supplemental information (Fig. S6) alongside the full script used in the field (in English and Malagasy).

We combined alternative levels of the four attributes in choice tasks using an efficient design that seeks to minimize the standard error of the coefficients to be estimated (Ferrini & Scarpa, 2007). The fractional factorial design was optimized for d-efficiency for the multinomial logit model using Ngene 1.1.1, and based on information on the signs of the parameters obtained from the piloting (Scarpa & Rose, 2008). The main purpose of this design was to ensure more reliable parameter estimates despite the relatively small sample size that was achievable given the field conditions (Rose & Bliemer, 2013). The design generated 12 choice tasks which were divided into two blocks; each respondent was presented with six choice tasks. Respondents were randomly assigned to one of the two blocks in the experiment.

Research ethics

The Bangor University College of Natural Sciences ethics committee approved this study (on October 29, 2013), and all members of the survey team received ethics training (covering confidentiality and informed consent). When we introduced the project we gave selected households a leaflet explaining the aims of the research with photos and names of the research team and contact details. We explained that participation in the research was voluntary, that they could leave at any time, and that no information that could identify them would be shared with others. We gave a small gift of useful items such as cups, pens, school books, or cigarette lighters to a total value of 3,000 ariary (approximately US$1) after interviews in phases one and three as a gesture of appreciation for their time. The detailed surveys in phase two took a whole day and required the household head to take us around his land holdings, therefore we paid respondents the local daily wage rate of 5,000 ariary (approximately US$1.85).

Data analysis

Characterizing poverty: Poverty is a multidimensional concept (Alkire et al., 2015). We therefore used a range of poverty indicators selected for the rural Malagasy context (see Table 2). The indicators of poverty were analyzed using a principal component analysis (PCA) in the R psych package (Revelle, 2017) based on polychoric and polyserial correlations estimated in the R polycor package (Fox, 2010). Input variables to the correlation matrix were measures of household food security, house size, house quality, access to lighting and education level of the household head (see Table 2).

Table 2 Key socio-economic characteristics of the surveyed households.

Variables	Description	Summary statistics	Coding for use in combined poverty index	
Number of rooms	Total number of rooms (including external kitchens)	Median = 2, Mean = 1.90, Std. dev. = 0.96	Continuous variable (0–10)	
House quality	Type of roof in the primary dwelling	77% thatch	Roof type (sheet metal = 2; thatch = 1)	
Food security (n = 452)	Number of months for which HH has enough to eat	Median = 7, Mean = 6.7, Std. dev. = 2.93	Continuous variable (0–12)	
Tropical livestock units	Numeric variable indicating the total livestock ownership of a household measured as “Tropical Livestock Unit” (107).	Median = 0.05, Mean = 0.42, Std. dev. = 1.15	Continuous variable (0–14.2)	
Irrigated rice	Whether the household has access to at least one irrigated rice field	63% no	0 = no; 1 = yes	
Access to lighting	Type of light and whether household have sufficient light	Median = 2, Mean = 1.96, Std. dev. = 1.05	Type of light (firewood = 0, candle, petrol or torch = 1; solar lamp or generator = 2) AND sufficient (never or rarely = 0, sometimes = 1, mostly or always = 2)	
‘Radio card’ mp3 player	Binary variable indicating whether the household has an mp3 device for playing music.	78% no	0 = no; 1 = yes	
Education level	Binary variable indicating low or high level of education of the household head. Low (0) = 0 to 5 years of schooling; High (1) = 6 or more years of schooling.	90% low	NA	
Household size	Total number of individuals considered members of the household.	Median = 5, Mean = 5.5, Std. dev. = 1.5	NA	
Ethnic group	Ethnic group to which the respondent household head belongs	Betsimisarika = 94%, Bezanozano = 3%
Other = 3%	NA	
Primary occupation n = 451	Main occupation of the household head	Agriculture = 90% Daily wage = 7% Other = 3%	NA	
Distance from the forest (km)	Distance of the household’s main home from the nearest protected area boundary (negative values refer to households based within the protected area).	Median = 2.1, Mean = 2.3, Std. dev. = 2	NA	
Household age (years) n = 441	The length of time a household has been established as an independent unit (since cohabiting or starting to farm independently)	Median = 10, Mean = 14.6, Std. dev. = 13	NA	
Note:

Where we don’t have valid data for the full data set of 453, the sample size is given in parentheses the first column. For variables included in our combined poverty index, we give details of how they were coded for inclusion in the PCA.

Converting ariary to US$: We used the World Bank’s Global Economic Monitor database on historic exchange rates to get US$—Malagasy ariary median exchange rate for the period of data collection between July 2014 and June 2015 (World Bank, 2017a). The median exchange rate thus obtained is 2,702 MGA for one US$, which is used in all our analyses and in conversion of local currency into US$. We use the seasonally adjusted consumer price index data from the same source to adjust any local currency values outside of the above period before converting to US$ figures for comparison.

Modelling choice experiment: The discrete choice data was analyzed with a mixed logit model using the R gmnl package (Sarrias & Daziano, 2015). The mixed logit approach introduces preference heterogeneity by “individualizing” preferences; each respondent has a possibly unique set of preference parameters (Train, 2003). As it is not practical to estimate the parameter vector governing the behavior of individual respondents, preference parameters are instead defined as random draws from a joint distribution and mixed logit models estimate a distribution of these parameters from the full sample. With the exception of the fixed payment parameter, all parameters are specified as random and given a normal distribution (truncated normal in the case of the opportunity cost parameter). The opportunity cost estimates were derived from the marginal rate of substitution between the forest clearance attribute and the monetary attribute. They are calculated as follows: Opportunity cost estimates=βiβprice

Where βi are the attribute coefficients of strict protection, and βprice are the price coefficients. Standard errors on the cost estimates were estimated from the mean and covariance matrix using the delta method.

Comparing the magnitude of opportunity costs with local incomes: We estimated household income for a subset of 102 households using the detailed surveys in phase 2 (see Supplemental Information). We annualized our estimates of opportunity cost using a 60 year time horizon and a range of discount rates of 0.001–5%. While selecting an appropriate time horizon and discount rates for annualizing NPV estimates is difficult, we argue that these choices are well supported and also conservative (as long time horizons and low discount rates result in lower annual costs). These calculations allowed us to present annualized opportunity costs as a percentage of the total household income.

Estimating the value of the compensation received by compensated households: We used a random card sort exercise (Shackley & Dixon, 2014) to help respondents estimate the value of the compensation provided by trading it off against seven levels of hypothetical cash payment. This elicited upper and lower bounds for respondents’ willingness to accept cash in place of the compensation. We then asked a single open ended contingent valuation question to elicit a single willingness to accept value (which always lay within the bounds identified by the random card sort, see Supplemental Information). This valuation was conducted ex-post to value the compensation as it had actually been delivered. Respondents were invited to take into consideration what they knew about how well the compensation project had worked and decide whether, if offered the opportunity to choose, they would choose the project or a cash sum. Debriefing questions found that 57 of 62 respondents felt the exercise “definitely showed” the true value of the project to them, suggesting this method was successful.

Estimating the total number of households bearing significant opportunity costs in CAZ: The median distance of our surveyed households from the protected area boundary was 2 km. Therefore we drew a conservative 2 km buffer around the outer boundary of the CAZ protected area (excluding other parks and reserves) to define the area for which we have information on local opportunity costs (Fig. S4). We ran the EcoEngine population algorithm in WaterWorld (Mulligan, 2013) using two spatial population datasets (1) Fokontany-level (Fkt2010) data (provided by the national statistics agency INSTAT Madagascar), and (2) Landscan 2007 (LS07) dataset (Bright et al., 2008) and masked to the shapefile of CAZ with the 2 km buffer to get population distribution within the area of interest (Fig. S4). We compared the population distribution from these two datasets to the census data we collected from our study sites (Fig. S5). The LS07 population estimates were much more representative of our census estimates. Using this data we estimated the population in and around the CAZ new protected area was 49,183 people at 1 km resolution grid, equating to 9,837 households with a median household size of five (see Table 2). We then multiplied this figure by our estimated proportion of households in the CAZ sites (site 1 and site 2) with NPV of opportunity costs higher than a range of thresholds to estimate the total number of households with significant opportunity costs for the whole of CAZ.

Estimating the global value of conservation of the CAZ: The CAZ protected area is projected to avoid the release of appropriately 1 million tonnes of CO2 per year for 10 years (Rainforest Alliance, 2013). We used this figure, and the average social cost per ton of CO2 for 2015 at 5% discount rate, US$ 11 (Interagency Working Group on Social Cost of Carbon, 2016), to estimate the social value of the CAZ protected area in terms of its contribution to climate mitigation as approximately US$110,000,000.

Results

Livelihoods of people in the CAZ

People living around the CAZ forest are extremely poor (Table 2). Food security is low: the median number of months for which families have enough to eat was just seven. Household assets are low: the median household owns just 0.05 Tropical Livestock Units (equivalent to five chickens). The vast majority of people live in small, poor quality houses of just one or two rooms made of local materials and have insufficient access to light (Table 2). Most household heads are illiterate or have less than two years of schooling. A total of 90% percent of people in the study area are dependent on swidden agriculture for their livelihood (Table 2). A total of 20% of respondents have obtained land directly from clearing the forest (others have bought or inherited cleared land), although this varies between sites (Fig. S1). Only 37% of households have access to irrigated rice fields (Table 2). PCA of seven measures of wealth resulted in two axes which explained 45% of the variation and revealed no systematic differences between our four sites in terms of wealth (Fig. 2). These two axes were used as covariates in analyzing the choice experiment.

Figure 2 Indicators of wealth.

Principal Component Analysis plots showing (A) loadings of measures of wealth, and (B) individual household scores with a convex hull for each site. Wealth axis 1 can be interpreted as an overall measure of wealth (a higher value indicates higher household wealth while wealth axis 2 distinguishes between households with larger, higher quality houses and those growing irrigated rice and with high animal numbers.

The magnitude and distribution of local opportunity costs of preventing swidden agriculture

The median net present value of the opportunity cost across households in all sites is US$2,375. (Fig. 3A; see Table S2 for the coefficients from the choice experiment). The opportunity cost per household varies between sites. Interestingly, the site-level effect on the net opportunity costs estimate was greater (more negative) for the site that did not receive compensation than the site which was assessed by environmental and social safeguard assessment (World Bank, 2012) as eligible to receive compensation and where some households did receive compensation (Figs. 3A and 3B). Opportunity cost estimates from the sites adjoining long-established protected areas (where communities have experience of conservation restrictions) are higher on average than sites adjoining CAZ (where conservation restrictions are new) (Fig. 3A). This may reflect the effect of experience; i.e., they are better able to estimate the costs of conservation as have experienced the challenge of switching to livelihoods not based on swidden agriculture (Rakotonarivo et al., 2017).

Figure 3 The Net Present Value (NPV) of net household opportunity cost of conservation restrictions estimated from the choice experiment.

(A) The distribution of opportunity cost across the four study sites. (B) Coefficient plot showing the effect of study site, distance of a household from the forest frontier, household age, education of the household head and the two wealth axes on the estimated household opportunity cost.

Households further away from the forest frontier and with more educated household heads expected lower opportunity costs. There was no effect of either wealth axis on net opportunity costs (Fig. 3B), implying a higher proportional burden for poorer households.

Using detailed agricultural surveys we estimated total annual household incomes for a subset of households at sites 1, 2 and 3 (Fig. 4). For a range of realistic discount rates (0.001%, 2.5% and 5%) over a 60 year timeframe, we estimated annualized opportunity costs, which for median NPV were respectively US$40, $77 and $125 (Supplemental Information). It is important to note, however, that median income households do not necessarily bear median opportunity costs. Our estimates of annualized opportunity costs represented around 27% to 84% of the median total annual household income in these three sites for median income households (Fig. 4). This proportion is greater for the poorest households compared with those who are less poor; this finding was consistent across the range of discount rates (Fig. 4).

Figure 4 The relationship between annualized household opportunity cost (from the discrete choice model) as a proportion of household income, plotted against household income (in 2015 US$).

Median household income (US$233) is indicated with a dashed vertical line. Lines are locally smoothed (Loess) fits to the data for the individual discount rate.

We estimated the number of households who might be considered eligible for compensation using three thresholds (net present value of opportunity cost being greater than twice, three times and four times median annual income). This results in estimates of 6,274; 5,922 and 5,521 households in and around the new protected area which we argue should be considered eligible to receive compensation (Fig. 5A).

Figure 5 The underfunding of conservation compensation.

(A) A comparison between the number of households with NPV of opportunity costs greater than 2–4 times median annual household income, the number of households identified to receive compensation, and our estimate of the number fully compensated. (B) Comparison between the magnitude of the median net present value (NPV) of household opportunity cost (from our choice experiment), the maximum projected spend on compensation, and households’ ex post valuation of the compensation provided (from our contingent valuation). The inset puts these figures in the context of our estimate of the carbon value per household of the REDD+ project in CAZ. Orange bars represent results from our survey and analysis, blue bars represent data obtained from published reports about the safeguarding process.

We estimate that the present value of opportunity costs borne by local people due to the conservation restrictions imposed by the CAZ REDD+ pilot project are between 13 and 15 million US$ (5,521–6,274 households multiplied by the estimated median opportunity cost of US$ 2,375).

The magnitude and distribution of the compensation received

The environmental and social safeguard assessment of the CAZ new protected area initially identified 2,500 households as negatively affected by the protected area (World Bank, 2012). Using our conservative estimate of the number of households bearing NPV of opportunity costs higher than 2–4 times their median annual incomes (see above), we suggest that there are between 3,020 and 3,770 households unidentified for compensation (between 15,100 and 18,850 people). Therefore, even with our most conservative estimate of the number of households eligible for compensation, less than 50% of these were identified by the World Bank process (Supplemental Information). According to the final project implementation report of the World Bank regarding the implementation of safeguards around CAZ, 1,012 of the 2,500 households had yet to receive compensation by the end of 2015 (World Bank, 2016) so even this conservative number is likely to be an underestimate of the proportion of eligible households who have not received compensation.

The feasibility plan for the implementation of the social safeguard scheme surrounding CAZ (MEEF/SAPM/CI, 2013) suggests that approximately US$ 100–170 would be spent on each household eligible for compensation, excluding transaction costs. For the households who received this compensation the projects went ahead as planned (with some adaptations—for example, in site 1, technical farming support for irrigated rice was replaced with support for rainfed beans). Assuming that the total budgeted amount for direct compensation per household was spent, the amount spent per household on compensation was therefore within the range of our estimate of annual opportunity costs, however, it is important to note that these are one-off projects, with no further support budgeted for subsequent years.

Using a contingent valuation exercise two years after project delivery with all 62 households who received the compensation in site 1, we found that on average these households valued the projects that they had received at a net present value of US$ 79. This is of the same order of magnitude as the annual opportunity costs estimated by the safeguard assessment ($120) (World Bank, 2012) but considerably less than the net present value of the opportunity cost (median = $2,375) (Fig. 5B). For the majority of households studied, these compensation projects covered less than 5% of their opportunity costs while only a few households with very low opportunity costs were better compensated (maximum ∼45%) (Fig. S2). We therefore conclude that none of the households were fully compensated (Fig. 5A).

The total projected spend on compensating local communities was approximately US$250,000–$425,000 ($100 or $170 multiplied by the 2,500 households who were initially identified for compensation). However, this is likely to be a significant overestimate of the actual compensation spend, since by the end of the project at least 1,012 remained uncompensated and some households from the initial list had been dropped for other reasons such as relocation or migration from CAZ (World Bank, 2016). Yet even this projected spend is two orders of magnitude less than our estimate of the total local opportunity cost. The amount of compensation spent per household is much lower than the carbon value that the REDD+ project in CAZ is expected to deliver during the first 10 years. Our conservative value for the carbon emissions which could be avoided over the life of the project (US$ 110,000,000, see above) represents approximately US$ 11,000 per household for every household within CAZ or 2 km of its borders (Fig. 5B).

Discussion

We have demonstrated that some of the poorest people in one of the poorest countries in the world are bearing very high opportunity costs due to conservation restrictions. These costs, when annualized, are a significant proportion of annual incomes. This is realistic when we consider that these costs are incurred over many years and indeed several generations. Despite the common narrative among conservationists that benefits from swidden agriculture are very short-lived as soil fertility is rapidly lost, and therefore any costs of conservation can also only be short-lived, the evidence does not support this (Brand & Pfund, 1998; Nielsen, Mertz & Noweg, 2006; Mertz et al., 2009; Rerkasem et al., 2009; Ziegler et al., 2009). In our own study sites, many families have been farming the same land through swidden cultivation for well over 100 years. Studies on traditional swidden agricultural systems in Madagascar and in other parts of the world generally agree that long-fallow swidden systems can be sustainable in the long term (Dove, 1983; Jarosz, 1996; Kerkhoff & Sharma, 2006; Erni, 2015) and can compete with more intensive farming systems in terms of returns to labor (Dove, 1983; Oxby, 1985; Nielsen, Mertz & Noweg, 2006). Swidden agriculture can also be of lower risk than alternatives and therefore be particularly vital to the poorest who have few alternatives (Nielsen, Mertz & Noweg, 2006; Scales, 2014). Extensive qualitative debriefing shows that respondents did consider the varied and multiple influences and made meaningful trade-offs in the choice experiments (Rakotonarivo et al., 2017). Qualitative evidence (Rakotonarivo, 2016) shows that people took the long view when considering their responses, and also that some respondents expected cultural losses as well as more tangible costs, from enforced cessation of swidden agriculture. When annualized, our estimates of opportunity cost are close to the official estimate in the CAZ environmental and social safeguard document (World Bank, 2012) which used a very different approach. They are also comparable to an estimate of the annual opportunity cost incurred by rural Ugandan farmers of forgoing agriculture on forestlands (US$ 354/household/year; Bush et al., 2013). Finally, a number of us have met Malagasy farmers who have been jailed (a very serious punishment in a country where prison conditions are very severe; Roth, 2006) for clearing forest; demonstrating how strongly people rely on this activity and that conservation restrictions have a serious local cost.

Why do local costs of conservation matter?

Excluding local people from protected areas, or restricting their livelihood options within those areas without compensation has a number of problems. First, it ignores the rights of local communities to manage their land and natural resources; an environmentally unjust situation (Martin, McGuire & Sullivan, 2013; Mcdermott, Mahanty & Schreckenberg, 2013) and results in some of the world’s poorest people bearing costs to supply global environmental benefits. Second, there can be implications for the sustainability of the conservation intervention itself as uncompensated losses can result in antagonism or even retribution (Naughton-Treves, Holland & Brandon, 2005). Sustainable management of protected areas in countries like Madagascar, with political instability, weak governance and poor infrastructure, depends in part on the goodwill of local communities (Rasolofoson, Nielsen & Jones, 2018). Illicit mining and logging have caused significant degradation in many of Madagascar’s protected areas in recent years (Allnutt, Asner & Golden, 2013; Rakotomanana, Jenkins & Ratsimbazafy, 2013; Schwitzer et al., 2014). While local communities cannot prevent these incursions by themselves, their cooperation is vital to the success of conservation (Fritz-Vietta et al., 2011), yet this cooperation is unlikely if protected areas bring only costs.

What about the local benefits?

The majority of people in eastern Madagascar collect a wide range of wild-harvested products for subsistence use and trade (including building materials, fibers, famine foods Ratsimbazafy, Harada & Yamamura, 2011), and may experience other benefits of maintaining standing forest. Our choice experiment was designed to estimate net costs, taking account of all influences (positive or negative) on a respondent’s utility and qualitative debriefing suggests that respondents did consider both costs and benefits of conservation when formulating responses. Our method cannot distinguish between those who have a net positive utility for forest conservation and those who experience no opportunity cost (i.e., are neutral). A small number (less than 15%) of responding households have zero net costs, perhaps because they live relatively far from the forest, are not dependent on agriculture or are too old to clear new lands. Some of these households may derive net positive benefits from forest conservation (due to cultural reasons or because they perceive forest to be important for providing clean water or air). We are not in a position to estimate the magnitude of utility these people might get from forest conservation, but this does not affect our estimates of the number of households that expect net-negative costs or the magnitude of those costs.

Is compensation reaching the right people?

We conservatively estimate that less than 50% of those who should have been eligible for compensation were identified as eligible. In one of our study sites (site 2), no compensation was distributed at all, but our estimates show that opportunity costs at this site are at least comparable to those in the site where compensation was distributed (site 1). At the site level, previous work by our team has demonstrated that in site 1 (where compensation was distributed) those reached were not necessarily the most deserving but were those with better socio-economic and political status locally, and easier to access geographically (Poudyal et al., 2016b). Furthermore, while 2,500 households were initially identified to receive compensation (World Bank, 2012), by the end of 2015 the World Bank stated that 1,012 of these households had yet to receive compensation and some others (no number provided) had been excluded from the list due to migration, or unwillingness to take part in the compensation programme (World Bank, 2016).

Is compensation sufficient?

The average one-off spend on providing compensation to each household who received it was similar to the average annual opportunity cost of swidden agriculture estimated by the CAZ environmental and social safeguard document (World Bank, 2012). It is highly unlikely that annual benefits can be generated that are of similar magnitude to the initial investment, and our valuation of the compensation received (two years after it was distributed) confirmed that local people valued the projects, on average, at slightly less than they cost to deliver. The costs of conservation are likely to be felt over decades therefore even those people who received compensation are under-compensated relative to the costs they will incur. The number of households in the CAZ for whom the safeguard compensation fully compensated for their opportunity costs is therefore zero.

Some time in 2018, the World Bank will launch its new Environmental and Social Framework; the result of four years of consultation on the existing environmental and social safeguard policies (World Bank, 2017b). While the aspirations for what the social safeguards seek to achieve with respect to economically displaced persons remain clear and strong, some experts have raised concern that oversight will be weakened under the new framework as responsibility to ensure safeguards are met is shifting from the lender (the Bank) to the borrower (Passoni, Rosenbaum & Vermunt, 2018). Given our work shows that even with existing levels of oversight, projected affected persons are under-compensated, this is concerning.

Can forest conservation in low-income countries be achieved without the poorest bearing the costs?

As Madagascar develops, it can be expected that many people will choose to move away from swidden agriculture towards more intensive agricultural systems and to livelihoods not based directly on the land (Jones et al., 2018); a transition which has been seen in other parts of the world (Cramb et al., 2009; Schmidt-Vogt et al., 2009). The question is, how can the forests be protected during that transition and while Madagascar’s economic development continues to be slow and beset by regular political crises?

Although complex, we argue that, resolving land tenure in forested areas (including recognizing and respecting customary rights) is vital if effective conservation is to be achieved without poor local people bearing the cost. Indeed, resolving issues surrounding tenure of forested land (particularly mature tree fallows) could also benefit local people and forest conservation for two reasons. First, if local peoples’ rights over forest are legally recognized, it puts them in a stronger position to argue for effective compensation, reduces the possibility of a resource rush (Sunderlin et al., 2014; Rakotonarivo et al., 2018), and would ultimately reduce the transaction costs of negotiating fair compensation (Pham et al., 2013). Second, by undermining customary tenure, weakly enforced state ownership can increase deforestation rather than reduce it (Horning, 2005). For example, if local people cannot exclude others from clearing “their” tree fallows, this provides perverse incentives for such land to be cleared more often than would be optimum for the customary owner; resulting in shorter fallow cycles and land degradation.

There has been recent progress in Madagascar in formalizing tenure, with the establishment of land tenure offices at the commune level (although the process of issuing land certificates is still slow and coverage of land offices is patchy; Burnod, Andrianirina-Ratsialonana & Ravelomanantsoa, 2014; Widman, 2014). Unfortunately, there are two significant challenges to resolving the tenure of farmers on Madagascar’s forest frontier. First, although mature tree fallows are locally considered part of agricultural land, the current forest code does not allow formal tenure to be granted over such land as it is considered to be state land (Jones et al., 2018). Second, under the current tenure laws (Laws 2005-019 and 2006-031), those living within the border of protected areas, i.e., many of those considered in this study, are not eligible to formalize their tenure.

We estimate that the total local opportunity costs of conservation restrictions in the CAZ protected area are US$13–15 million, while the total amount projected to be spent on compensating households was less than US$425,000. This suggests that substantially greater investment in compensation is needed to ensure that local opportunity costs are compensated; greatly increasing the implementation costs of such projects. Because opportunity costs will be incurred over a long time period, this compensation could also be spread over several decades (although this is no excuse for complacency: costs will be felt by some households from the first years of protected area establishment). Global conservation efforts are already underfunded by at least an order of magnitude (McCarthy et al., 2012; Waldron et al., 2013). However, when put in the context of the global value of ecosystem services lost due to land use change (Costanza et al., 2014) and the fact that a conservative estimate of the value of CO2 emissions avoided by protecting the CAZ is over US$110 million over 10 years, the figures involved are relatively small. They would, however, require a major change in resource allocation to provide sufficient funds to compensate for opportunity costs (and cover the significant transaction costs associated with safeguard compensation programs; Mackinnon et al., 2017).

What are the implications of this work for the implementation of REDD+ social safeguards?

Following the 2015 Paris climate agreement, REDD+ was formally confirmed as part of the global tool kit for mitigating climate change. The UNFCC Cancun agreement (UNFCCC, 2011) had already laid out the safeguards that REDD+ programs must follow to avoid negative environmental or social impacts (Decision 1/CP.16). The Cancun safeguards are very different from the World Bank social safeguards and do not explicitly refer to compensation for affected persons, but they do require that knowledge and rights of local communities are respected, that there is effective local participation in REDD design and implementation, and social co-benefits are promoted. Madagascar is currently in the process of finalizing its national REDD+ strategy, which includes developing social and environmental safeguards in line with the Cancun commitments. We suggest that there are important lessons from our work to inform that strategy; especially given the paucity of published work exploring the effectiveness of REDD+ social safeguards (Duchelle et al., 2017). These are: there are significant and long lasting costs to local people from restrictions on clearing forest land for agriculture, these should be addressed both for environmental justice reasons but also to improve the sustainability of forest conservation and this will require significant investment. Reaching the poorest and most marginalized is difficult and deserves special attention. Finally, rigorous and independent monitoring will be needed to ensure that any safeguards program achieves its stated objectives on the ground.

Conclusions

Conservation as a movement recognizes that sustainable management of natural resources cannot be achieved without considering local people. This has resulted in very positive commitments to avoid negative impacts of conservation restrictions on local communities. However, there has been little formal scrutiny of the extent to which these commitments are delivered upon. We evaluate an example of a new protected area that has been established with commitments to avoid negative impacts on local people. Unfortunately we show that the local people, some of the poorest in the world, have lost out as a result of the protected area establishment, and that compensation provided to mitigate these costs has been inadequate. Too little has been received by too few and it has not reached those most in need. These are challenging results to present and we do so cautiously. We recognize that the individuals and organizations involved are often doing their best in very challenging circumstances. However, we want to draw global attention to the fact that having policies in place to protect local people from the costs of conservation is not sufficient: they must be accompanied by adequate investment, over long periods. There is no straightforward solution and effective compensation will be expensive. However we argue that ignoring the issue of local costs is both unjust (and therefore immoral) and also unsustainable. Real change and substantial new investment is needed.

Supplemental Information

Supplemental Information 1 Fig. S1. How households gain access to land in the study sites.

Y-axis shows the overall percentage of plots in each site being accessed through one of the five ways listed–total adding up to 100% for each site. Error bars show 95% confidence intervals.

Click here for additional data file.

Supplemental Information 2 Fig. S2. The value of the compensation projects as a percentage of household opportunity cost.

Data is for 62 recipients of compensation (all from site 1), 2 years after compensation was received. The value of compensation is estimated from our contingent valuation while the opportunity cost of conservation is estimated from the choice experiment.

Click here for additional data file.

Supplemental Information 3 Fig. S3. Pictures showing the context of the field work presented in this paper.

a), b) The biodiversity of the CAZ is world-renowned. c), d) 10s of 1000s of people live around the CAZ new protected area, traditionally most people depend economically on clearing land for agriculture in a swidden system known locally as ‘tavy.’ e) The CAZ protected area will result in strict enforcement of conservation rules including not clearing new land. f) Selected residents have been identified as Project Affected Person and therefore have received micro-development projects such as improved bean cultivation under World Bank safeguards. g) To build a sampling frame we worked with local leader to update available maps and then visited each hamlet with a GPS. h) Our team stayed in the villages (with local families) where worked for extended periods which facilitated trust.

Click here for additional data file.

Supplemental Information 4 Fig. S4. Population distribution within a 2 km buffer of CAZ.

Established protected areas managed by Madagascar National Parks (with 2 Km buffer around them) have been excluded as different compensation right exist there. The population model is based on Landscan 2007 data distributed with the EcoEngine algorithm in WaterWorld.

Click here for additional data file.

Supplemental Information 5 Fig. S5. Examining goodness of fit of the modelled population from existing sources with our population data collected from the study areas.

Our field data shows the primary census data collected in each fokontany during 2014/2015 (population data from p4ges field sites) plotted against the population estimates for those sites from LandScan 2007 (LS07), and INSTAT (Madagascar’s National Institute of Statistics) 2010 data (fokontany 2010)- both distributed using EcoEngine algorithm in WaterWorld.

Click here for additional data file.

Supplemental Information 6 Fig. S6. Example choice card of the Discrete Choice Experiment used to estimate opportunity costs.

Click here for additional data file.

Supplemental Information 7 Table S1. Attributes and levels of the choice experiment (reference levels in bold).

Click here for additional data file.

Supplemental Information 8 Table S2. The coefficients from the choice experiment. Lower and upper bounds are 95% confidence intervals.

Click here for additional data file.

Supplemental Information 9 Table S3. The annualised opportunity costs of conservation per household (USD).

Click here for additional data file.

Supplemental Information 10 Survey instruments for phase one of field work.

(household survey and choice experiment in English and Malagasy).

Click here for additional data file.

Supplemental Information 11 Survey instruments for phase two of field work.

(Agricultural survey in English and Malagasy).

Click here for additional data file.

Supplemental Information 12 Survey instruments for phase three of field work.

Contingent valuation of the compensation recieved by housholds (in English and Malagasy).

Click here for additional data file.

We thank local leaders and all the people interviewed for participating in this research. We also thank the Malagasy government for research permission. We thank D.T. Rafanomezantsoa, R. Heriniaina, T. Razafimandimby, and T. Brodin for help with the fieldwork and J. Jacobsen for support with the choice experiment. We thank Conservation International Madagascar, K. Schreckenberg, P. Ranjatson and the wider p4ges team for useful discussion. M. Mulligan helped with our population estimates. The World Bank Madagascar shared information on the safeguarding process in CAZ. We thank the Malagasy government’s national coordination office of REDD+ (particularly M. Andriamanjato) for discussion about the emerging social safeguard strategy for REDD+.

Additional Information and Declarations

Competing Interests

Author Contributions

Human Ethics

Field Study Permissions

Data Availability

The authors declare that they have no competing interests.

Mahesh Poudyal conceived and designed the experiments, performed the experiments, analyzed the data, prepared figures and/or tables, authored or reviewed drafts of the paper, approved the final draft.

Julia P.G. Jones conceived and designed the experiments, performed the experiments, analyzed the data, prepared figures and/or tables, authored or reviewed drafts of the paper, approved the final draft.

Sarobidy O. Rakotonarivo conceived and designed the experiments, performed the experiments, analyzed the data, prepared figures and/or tables, authored or reviewed drafts of the paper, approved the final draft.

Neal Hockley conceived and designed the experiments, performed the experiments, analyzed the data, prepared figures and/or tables, authored or reviewed drafts of the paper, approved the final draft.

James M. Gibbons conceived and designed the experiments, analyzed the data, prepared figures and/or tables, authored or reviewed drafts of the paper, approved the final draft.

Rina Mandimbiniaina performed the experiments, approved the final draft.

Alexandra Rasoamanana performed the experiments, approved the final draft.

Nilsen S. Andrianantenaina performed the experiments, approved the final draft.

Bruno S. Ramamonjisoa authored or reviewed drafts of the paper, approved the final draft.

The following information was supplied relating to ethical approvals (i.e., approving body and any reference numbers):

Ethical approval was provided under Bangor University’s research ethics framework. Approval was granted on October 29, 2013.

The following information was supplied relating to field study approvals (i.e., approving body and any reference numbers):

Research permission was granted by the Ministry of Environment, Ecology and Forests (45/14/MEF/SG/DGP/DCB.SAP/SCB).

The following information was supplied regarding data availability:

GitHub: https://github.com/mpoudyal/cepaper.

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
