# Peer review of "Who bears the cost of forest conservation?"

_PeerJ, doi:10.7717/peerj.5106_

## Round 0.1 · original submission · Minor Revisions

The reviewers agree that this is an important paper, but relatively minor revisions are required before it can be accepted.

Reviewer 1 ·

Basic reporting

Ok here

Experimental design

OK here

Validity of the findings

OK here

Additional comments

Please see attached .pdf file with comments for both the author and editor.

Annotated reviews are not available for download in order to protect the identity of reviewers who chose to remain anonymous.

Reviewer 2 ·

Basic reporting

No comment

Experimental design

No comment

Validity of the findings

No comment

Additional comments

I found this article to be well written and the methods clearly explained. I noted that the choice experiment design had a problem with the lack of a 'status quo' option, due to data sensitivities and legal issues. This is a matter of some debate in the literature with purists claiming that this omission means that welfare estimates cannot be found. However, I think the second best approach used by the authors and the qualitative follow checks are reasonably convincing. The other feature of the valuation exercise which has also been the topic of much debate in the literature is WTBC versus WTP, the choice of WTBC will have a bias effect on the results.
Overall the authors have adopted mostly 'conservative' assumptions and have gone to great lengths to collect meaningful data sets.
I also note that the authors say their local costs results are in line with a study from Uganda (lines 477-479). A recently published study on Tanzania has the following local costs: $191-668 /ha, see J.Green et al (2018) Local costs of conservation exceed those borne by the global majority, Global Ecology and Conservation.

·

Basic reporting

No comment.

Experimental design

Given the reference to qualitative data and the importance of this in your interpretation, I wondered why there wasn’t more detail given in the methods and results. I understand that these are referenced in other papers (Rakotonarivo et al) but isn’t that also the case for the quantitative data?

Validity of the findings

The standards are met but I have suggested some clarification below regarding the choice experiment results.

Additional comments

This is an really important piece of work that is the only case I know of that compares compensation provided to conservation costs experienced by local people. It show that costs are substantially higher than gains over the longer term, especially for the poorest – with implications for justice and sustainability.

I am generally skeptical of economic methods such as contingent valuation applied cross-culturally in rural settings. These methods need to be underpinned by substantial local experience, qualitative research and extensive piloting in order to produce valid data and to understand what the answers people give really mean. I am convinced that is what has been undertaken here and there appears to be a meticulous attention to detail in applying these methods, but I have one query with regard this:

The phrase ‘opportunity costs’ seems to be glossing over a lot of issues that I think it needs clarification, especially in light of some of the criticisms of WTP/A methods. Perhaps your qualitative data can provide this. Do the high estimated costs encompass non-material losses for example the cultural attachment to swidden farming, a sense of autonomy/control etc (alongside cultural reasons for valuing conservation as suggested in line 516)? The fact that those with more experience of forest conservation have higher opportunity costs (lines 384-388) suggests that they understand the full extent of the range of costs, but also perhaps shows resistance to further loss of rights. In the introduction (line 99), you seem to suggest that this method does not capture multi-dimensional wellbeing, but it’s not clear what the respondents' thought processes are here. If a range of non-tangibles are captured, it suggests that perhaps monetary/material compensation is not simply commensurable – and seems to be in keeping with your conclusion that land tenure (rather than or in addition to compensation) is the solution, which otherwise isn't fully justified.

---

## Round 0.2 · accepted · Accept

I confirm that you have adequately addressed the reviewers concerns and the paper is now ready for publication

#